# Complications in Spinal Fusion Surgery: A Systematic Review of Clinically Used Cages

**DOI:** 10.3390/jcm11216279

**Published:** 2022-10-25

**Authors:** Francesca Veronesi, Maria Sartori, Cristiana Griffoni, Marcelo Valacco, Giuseppe Tedesco, Paolo Francesco Davassi, Alessandro Gasbarrini, Milena Fini, Giovanni Barbanti Brodano

**Affiliations:** 1Surgical Sciences and Technologies, IRCCS-Istituto Ortopedico Rizzoli, 40136 Bologna, Italy; 2Department of Spine Surgery, IRCCS-Istituto Ortopedico Rizzoli, via di Barbiano 1/10, 40136 Bologna, Italy; 3Department of Orthopedic and Traumatology, Hospital Churruca Visca, Buenos Aires 1437, Argentina; 4Scientific Direction, IRCCS-Istituto Ortopedico Rizzoli, 40136 Bologna, Italy

**Keywords:** spinal fusion, spinal diseases, cages, complications, biomaterials

## Abstract

Spinal fusion (SF) comprises surgical procedures for several pathologies that affect different spinal levels, and different cages are employed in SF surgery. Few clinical studies highlight the role of cages in complications beyond the outcomes. The aim of this systematic review is to collect the last 10 years’ worth of clinical studies that include cages in SF surgery, focusing on complications. Three databases are employed, and 21 clinical studies are included. The most-performed SF procedure was anterior cervical discectomy and fusion (ACDF), followed by lumbar SF. The polyetheretherketone (PEEK) cage was the most-used, and it was usually associated with autograft or calcium phosphate ceramics (hydroxyapatite (HA) and tricalcium phosphate (βTCP)). For lumbar SF procedures, the highest percentages of subsidence and pseudoarthrosis were observed with PEEK filled with bone morphogenetic protein 2 (BMP2) and βTCP. For ACDF procedures, PEEK filled with autograft showed the highest percentages of subsidence and pseudoarthrosis. Most studies highlighted the role of surgical techniques in patient complications. There are many interacting events that contextually affect the rate of clinical success or failure. Therefore, in future clinical studies, attention should focus on cages to improve knowledge of chemical, biological and topographical characteristics to improve bone growth and to counteract complications such as cage loosening or breaking and infections.

## 1. Introduction

Spinal fusion (SF) is one of the most common surgical procedures for treating conditions of the spine, including deformity, trauma, degenerative disc disease (DDD), spondylolisthesis and tumors, where removal of the damaged anatomical structure is required [1]. The removal of pathological tissues results in spine mechanical instability, whereas the main goal of SF surgery is to fuse two or more vertebras by inducing bone growth between the segments. Various techniques have been reported to achieve adequate bone healing and solid fusion, with different surgical approaches, graft materials and instrumentation. Due to the increase in complex surgical interventions following traumatic events and oncological and degenerative diseases linked to the aging of the population, research into new techniques and materials to improve the SF surgery success rate and reduce the percentage of pseudoarthrosis is strongly increasing.

Iliac crest autologous bone graft (ABG) is the “gold standard” for SF because of its osteoconductive, osteoinductive, and osteogenic properties combined with a microarchitecture that facilitates cell migration, proliferation and tissue regeneration. However, ABG is restricted by the limited supply and associated possible complications [2]. These aspects have led to an increase in the development and use of bone graft substitutes and biological agents. Local autografts, allografts, demineralized bone matrix (DBM), bone morphogenetic proteins (BMPs), autogenous growth factors (platelet derivatives), bone marrow aspirate (BMA), mesenchymal stem cells (MSCs) and synthetic bone grafts (ceramics) are increasing in popularity and use in SF procedures [3,4]. In parallel with the search for new bone graft substitutes for SF, the focus of spinal research since the 1800s has been on finding the perfect cage for spinal implantation. The cages used in spinal surgery devices have undergone a constant evolution as knowledge of the biomechanical principles of spinal instability has increased and new technologies and materials have become available for device manufacture [5].

Spinal implants need to demonstrate biostability (ability to resist the effects of pathological microorganisms) and biocompatibility as well as appropriate biomechanical characteristics (i.e., Young’s modulus, stiffness, fatigue and tensile strength) with few artifacts on imaging [6]. Among implants used, cages are devices that act as stabilizers for force distribution between vertebral bodies and restore the height of the intervertebral and foramina space. They allow vertebrae to fuse and heal when an intervertebral disc has failed [7,8]. Cages are typically made of metal (ranging from pure titanium (Ti) to titanium composite/alloy), ceramic (usually silicon nitride), or plastic (usually polyetheretherketone (PEEK) or another bioinert plastic such as acrylic), by itself or coated with another material (such as hydroxyapatite (HA) or titanium). Cage porosity allows bone growth and stabilization [9]. The most popular materials used are titanium alloys (titanium–aluminum–vanadium (Ti6Al4V)) and PEEK [10]. Ti alloys are the preferred metal in orthopedic implants due to their high fracture resistance and biocompatibility. The major issue with Ti alloys is its low bone-bonding ability; thus, increased research has been done on surface chemical and physical and morphology modifications to improve bone bonding [11,12]. PEEK provides stability similar to that of Ti alloys, and in some cases improves durability, strength and overall biomechanical profile. PEEK shows physiological load-sharing and low stress at the interface with the bone, with a reduction in the likelihood of adjacent degeneration, vertebral body bone loss and/or screw loosening [13,14]. PEEK has radiographic properties that allow surgeons to better monitor possible migration and the success of the implant. However, the primary issue with PEEK is that it is hydrophobic and unable to sufficiently bond to bone to achieve solid fusion. This may be associated with cage migration and pseudarthrosis.

In the literature, several reviews have already listed cages for SF procedures with their therapeutic potential [9,15]. Most studies that compare different cages in spinal surgery focus on biomaterial biocompatibility and physical properties, neglecting the probability of patient complications. As in all surgical procedures, complications can arise that are obviously related to the surgical procedure itself and do not depend on the cage used. However, for other types of reported complications in the context of SF, the connection with the type of cage used cannot be ruled out. It is reported that pseudoarthrosis (failed fusion) ranges from 2–30% at cervical sites [16], and cage subsidence ranges from 16% to 70% [17].

Thus, the aim of this review is to summarize and understand if and which type of cages used in SF procedures are possibly connected to complications.

To the best of the authors’ knowledge, a detailed review of all the complications associated with the cages used for SF has never been reported. The authors are aware that it is an ambitious aim and that the factors involved in the onset of spinal surgery complications are many and diverse: the surgical approach, the surgeon’s experience, the patient’s clinical conditions, the cage’s properties, and the clinical needs. Identifying whether the complication is linked to the cage used is very difficult, but with this review, we want to try to understand this aspect by reviewing clinical studies in the last ten years focused on lumbar and cervical SF.

## 2. Materials and Methods

### 2.1. Eligibility Criteria

A PICO question (Population of interest (P), Intervention (I), Comparators and Outcomes (CO)) statement was formulated to select and analyze only the relevant papers.

The Population considered was clinical studies in which patients were affected by degenerative spinal diseases. Randomized, prospective, retrospective and observational clinical studies were included. The Intervention was SF procedures with specific indication of any type of cage used.

The Comparator was any reference group.

The considered primary Outcome was reported complications associated with the SF procedures to understand if a relation can be established to the cages used. In addition, a secondary outcome was the radiological investigation performed to assess the success of fusion.

### 2.2. Search Strategy

The search was performed on 4 January 2021 and included research published from 1 January 2011 to 1 January 2021 according to the Preferred Reporting Items for Systematic Reviews and Meta-Analyses (PRISMA) statement (Figure 1 and Table 1). The search was carried out on three electronic databases (PubMed, Scopus and Web of Science) to identify relevant papers using the following keywords with Boolean operators: “(Scaffolds OR cages OR scaffold OR cage) AND (spinal fusion)”.

The limits were:(1)In PubMed: (i) types of papers (Clinical Study; Clinical Trial; Clinical Trial, Phase I; Clinical Trial, Phase II; Clinical Trial, Phase III; Clinical Trial, Phase IV; Comparative Study; Multicenter Study; Randomized Controlled Trial); (ii) language (English); and (iii) publication date (from 1 January 2011 to 1 January 2021),(2)In Scopus and Web of Science: (i) language (English); (ii) publication date (between 2011 and 2021); and (iii) types of papers (articles).

Relevant articles were screened using the title and abstract by two reviewers (FV and PD), and articles that did not meet the inclusion criteria were excluded. Only clinical studies evaluating complications related to the use of cages employed in SF surgery were included in this review; articles were submitted to a public reference manager to eliminate duplicates and to manage the references.

### 2.3. Information Extracted from Articles

The included full-text articles were retrieved and reviewed by the two reviewers, and any disagreement was resolved through discussion until consensus was reached or with the involvement of a third reviewer (MF). The researchers involved in the process of reviewing the papers used an Excel spreadsheet to independently perform the screening and data extraction. The following information was extracted from each paper to summarize the evidence reported in each study: (a) type of study and follow-up (f-up), (b) type of cages implanted, (c) inclusion criteria and patient allocation, (d) reported complications, and (e) results, mainly related to the radiological outcome in terms of SF.

### 2.4. Risk of Bias Assessment

Researchers also evaluated the risk of bias of the records included in the review in accordance with the Coleman methodology score (Figure 2) [18]. Any disagreements were resolved through discussion.

## 3. Results

The initial literature search retrieved 144 studies from PubMed, 277 from Scopus and 261 from Web of Science, for a total of 682 articles. There were 585 identified papers after duplicates (97 records) were removal with Mendeley software. After screening the titles and the abstracts, 237 articles were obtained. Reviews, in vivo, in vitro or ex vivo studies, case reports, and studies regarding biomechanical or mathematical models or finite elements were excluded, for a total of 348 articles. Among them, 119 studies were excluded because they did not evaluate and report complications. The remaining 118 articles were considered eligible. After reading the full texts, a total of 21 articles were included in this systematic review in agreement with the PICO question and the PRISMA methodological tool (Figure 1).

The studies were grouped based on the surgical procedures used:

(1) Lumbar SF, such as posterolateral lumbar (PLF), posterolateral lumbar interbody fusion (PLIF), anterior lumbar interbody fusion (ALIF), transforaminal lumbar interbody fusion (TLIF) and extreme lateral interbody fusion (XLIF) in 9 studies;

(2) Anterior cervical discectomy and fusion (ACDF) in 12 studies.

Table 2 summarizes the main information extracted from the included clinical studies according to the PICO question.

### 3.1. Lumbar SF

In nine studies, PEEK-based cages were used for lumbar SF procedures (one- or multi-level PLIF, TLIF, ALIF and XLIF). Three studies [19,20,22] compared the use of PEEK cages with different types of cages: an ABG obtained from spinous process and laminae [19], a Biocage obtained from allogeneic cortical bone [20] and a Ti alloy cage (Ti cage + ABG) [22]. Sixty-nine, 379 and 34 patients underwent PLIF [19,20,21], while 117, 22 and 40 patients were submitted to TLIF [22,23,24] procedures. PEEK cages were also used prefilled with β tricalcium phosphate (βTCP) and impregnated with iliac crest BMA or HA (PEEK + βTCP + BMA cage and PEEK + βTCP + HA cage, respectively) [21,25,27]. Additionally, rhBMP2 solution applied to an absorbable collagen sponge (ACS) was added to the PEEK cage (PEEK + rhBMP-2 cage). In these studies, 110, 131 and 84 patients were treated with XLIF, ALIF and ACDF, respectively [26,27,37]. Twenty-two patients underwent PLIF or TLIF procedures with dynamic posterior pedicle screw/rod-based stabilization performed with the dynamic part of PEEK and silicon and the pedicle screws of standard Ti alloy (PEEK cage + silicon + Ti screw) [23], and an autograft mixed with HA and β-TCP (65/35) and a PEEK bullet-shaped oblique cage with and without Ti alloy coating (TiPEEK cage) was employed in 40 patients in one- or two- level ALIF or TLIF procedures [24].

### 3.2. Complications

After 24 months, PEEK cages showed complication rates comparable to those of ABG, with no surgery-related neurological deficit, wound breakdown, or hardware loosening or breakage. In the PEEK cage group, dural tears and superficial wound infections were very low [19]. The complication rates observed in the PEEK + ABG group after 32 months were like those observed in Biocage patients and were very low, among them were low pseudoarthrosis (5.2% PEEK + ABG group vs. 3.4% Biocage) and subsidence (1.7% PEEK + ABG vs. 1.9% Biocage) [20]. No infection of the surgical site was observed during a 24-month f-up period in patients with PEEK cages and Ti alloy cages, both filled with ABG. In addition, pseudoarthrosis was observed at 10 levels with PEEK + ABG cages and at 16 levels with Ti + ABG cages [22]. The pseudoarthrosis rate was 26.5% when PEEK + βTCP + BMA cages were used in PLIF procedures [21], while in ALIF procedures, the pseudoarthrosis rate was 10% after 12 months [25]. PEEK + rhBMP-2 cages showed subsidence in 11% and pseudoarthrosis in 6% of cases with very few other complications after 24 and 12 months [26,27]. PEEK cage + silicon + Ti screw led to a high rate of implant failure and adjacent segment degeneration (18% and 15%, respectively) at 24 months. Radiculopathy, misplaced pedicle screw, superficial wound infection, pulmonary disease and incidental durometry were very low and were present in one patient for each method [23]. TiPEEK cages appeared to have no negative effects on outcome or safety in the short term. Pseudoarthrosis was comparable to that of the PEEK cage (10% in both cages), with low pedicle screw loss, persistent leg pain, hematoma compressing a nerve root and post-operative wound infection, all related to the procedure rather than to the cages, and comparable between the two groups at 12 months [24].

### 3.3. Clinical Outcomes

Between 8 and 12 months, fusion occurred in 94.1% of patients with PEEK cages and in 97.1% of patients with ABG. All remaining patients achieved successful fusion by 24 months, and there was no significant difference in fusion rates between the two groups. Pain reduction and improvement of functional outcomes were obtained in both groups [19]. Wu et al. compared the safety and efficacy of a Biocage with that of a PEEK + ABG cage, and no significant differences were found in the fusion rate, pain reduction and improvement of functional outcomes. During f-up, the mean intervertebral space height and intervertebral foramen height recovered significantly in the Biocage group compared to the PEEK + ABG cage group [20]. Tanida et al. compared PEEK + ABG cages and Ti cages + ABG, and concluded that the bone union rate did not differ significantly between the two groups (80.4% PEEK + ABG vs. 82.8% Ti cage + ABG) [22]. When PEEK + βTCP + BMA cages were used, good clinical results were obtained, and the average Oswestry Disability Index (ODI) and Visual Analogue Scale (VAS) for leg and back and fusion rate (47.7% and 85.5%, respectively) improved significantly [22,23]. When PEEK + rhBMP-2 cages were used, a reduction in ODI and VAS and an increase in Physical Component Summary (PCS) and Mental Health Component Summary (MCS) scores was observed [21,25], along with a higher fusion rate compared to PEEK + βTCP + HA cages (96% PEEK + rhBMP-2 cages vs. 80% PEEK + βTCP + HA cages) [27]. PEEK cage + silicon + Ti screw had a reduced Core Outcome Measures Index (COMI) and VAS [23]. The rate of complete or partial fusion of TiPEEK cages at 3 months was 91.7% Overall, there were no significant differences in ODI or in radiological outcomes between PEEK and TiPEEK cages after 12 months [24].

### 3.4. Anterior Cervical Discectomy and Fusion

Twelve studies regarded ACDF procedures. Among them, eight studies used cages made from PEEK filled with bone in 60, 28, 54, 100, 98, 68 and 47 patients undergoing one- or two-level ACDF [28,29,30,31,32,33,34,35]. Cages were also made by combining PEEK and βTCP or rhBMP2 in 64 and 191 patients, respectively [36,37]. A ridged Ti alloy endplate combined with a PEEK body and allograft (PEEK + Ti alloy) formed the cages used in 25 patients [38]. One study used Ti alloy cages instead of PEEK cages, and the authors compared Plasmapore-coated Ti alloy cages with non-coated cages in 72 patients [39].

### 3.5. Complications

PEEK cages filled with local bone graft (PEEK + ABG) did not show complications. The only use of ABG showed a low complication rate, donor site chronic pain, surgical wound infection and reoperation due to a broken fixation system screw, underlying that 83.3% of complications were inherent to bone graft harvesting from the iliac crest and not to the cages. Pseudoarthrosis was not observed in all patients [28,29]. Donor site pain and wound infection were observed only in the ABG group, and lower dysphagia was only in the PEEK + ABG group after 24 months [29]. PEEK + ABG did not evoke intraoperative complications, neurological deterioration or wound infections, and the operation time and bleeding were significantly higher with four-level ACDF than with two- or three-level ACDF. Subsidence was similar in all ACDF cases (10% in two-level, 20% in three-level and 25% in four-level), and pseudoarthrosis was observed in two- and three-level ACDF (10%) at 24 months [30]. ACDF performed using self-locking, stand-alone cages filled with bioceramic artificial bone showed no perioperative cerebral fluid leakage, wound infection, hematoma, cage migration or plate-related complications. Compared to ACDF using cages and plate fixation, the pseudoarthrosis rate was similar (4.3% without plate fixation vs. 7.7% with plate fixation) with a higher subsidence rate not related to the cage used (7.1%) after 29 months [31]. In the other four studies, PEEK + ABG cages filled with ABG composed of osteophyte autograft [32], morselized bone from the local decompression [33] or autogenous iliac cancellous bone [34,35] were compared to a cage made of silicon nitride spacers filled with microporous silicon nitride (silicon nitride + blood cage) [32], nano-hydroxyapatite/polyamide66 filled with morselized bone from the local decompression (nHA/PA66 + ABG cage) [33] and PEEK cages filled with calcium sulphate/demineralized bone matrix pellets (PEEK + CS/DBM cage) [34] or filled with PolyBone, a βTCP material designed to act as a substitute bone graft (PEEK + PolyBone cage) [35]. PEEK + ABG cage showed similar operation time, blood loss, length of stay (LOS), subsidence, transient dysphagia, incidental durotomies and recurrent symptomatic nerve root compression compared to silicon nitride + blood cage, but there were fewer revision surgeries at the adjacent level (6.25% in the PEEK + ABG group vs. 11.54% in the silicon nitride + blood one). The authors did not specify the reason subsidence occurred in one patient in each group at 24 months [32]. There were no allergic reactions with n-HA/PA66 and no neurological damage, but there was similar wound infection and subsidence compared to PEEK + ABG (9.8% PEEK + ABG vs. 10.6% n-HA/PA66) at 96.4 months. The authors supposed that subsidence might primarily be due to the unsuitable shape of the cages and the occasional need for the cage to be cut intraoperatively to obtain a better shape [33]. In addition, PEEK + ABG showed higher operation time, blood loss and total complication rate compared to PEEK + CS/DBM, but the related complications were not due to the cage, and no additional surgery was performed in any case. Furthermore, no device-related complications, such as hardware loosening and/or breakage, screw pullout, or displacement of the cage, were observed in either group at 24 months [34]. Lastly, the same PEEK + ABG cage showed no surgically related complications, similar to PEEK + PolyBone, at 30 months [35]. Like acrylic cages, which are made of polymethylmethacrylate (PMMA), PEEK + βTCP cages did not evoke neurological deficits, wound infections, cerebrospinal fluid fistulas, direct damage to the esophagus or trachea, or hemorrhages requiring transfusion, even if new degenerative changes at each level of the cervical spine (25% with acrylic cages vs. 34.4% with PEEK + βTCP cages), transient hoarseness and disk herniation at the lower level were similar in both groups and were low at 12 months [36]. In addition, PEEK + βTCP cages reduced 30-day readmissions and oral steroids use and increased subsequent cervical spine surgery more than PEEK + rhBMP2. Mild hardware failure showed with PEEK + βTCP cages (1.87%) at 12 months [37]. PEEK + Ti alloy showed 9.10% pseudoarthrosis, in which bridging of bone occurred outside the implant. However, the patients experienced good clinical outcomes, and there were no implant-related complications, implant failures, post-operative hematomas or infections after 14.6 months [38]. Instead of PEEK cages, Takeuchi et al. used a cage coated with Plasmapore Ti; it showed no cage migration, infection, or complications related to the surgery at 24 months [39].

### 3.6. Clinical Outcomes

The use of ABG and PEEK + ABG cages showed an improvement in clinical results, Japanese Orthopaedic Association (JOA) score, fusion rate and disc space height, and a reduction in VAS over time, with a similar fusion rate (95.2% in PEEK + ABG group vs. 95.7% in the ABG group in one study and 93.1% in PEEK + ABG group vs. 90.3% in the ABG group in another study) [28,29]. PEEK + ABG reduced VAS and increased solid fusion (92.85%) over time, regardless of the operated levels (2-, 3- or 4- level ACDF) [30]. Self-locking PEEK + bioceramic artificial bone cage showed an increase in the Neck Disability Index (NDI) and the JOA score over time [31]. When comparing PEEK + ABG and silicon nitride + blood, the two cages showed a similar increase in the NDI, SF36, patient perceived recovery and fusion rate over time, with reduction in VAS [32]. Like n-HA/PA66 + ABG, PEEK + ABG increased the fusion rate (97.8% in the PEEK + ABG group vs. 98.1% in the n-HA/PA66 + ABG group), segmental lordosis and the JOA score over time and reduced the VAS score [33]. The fusion rate was also similar between PEEK + ABG and PEEK + CS/DBM cages (100% at final follow-up) [34]. PEEK + PolyBone cages showed a similar reduction in NDI and Numeric Pain Rating Scale (NRS) scores and an increase in fusion rate compared to PEEK + ABG cages, but with lower disc height and greater time taken for fusion in patients treated with PEEK + PolyBone cages [35]. With PEEK + βTCP cages, clinical outcome was worse, and there was reduced disc space height and fusion rate, more so than with acrylic cage (93.8% vs. 96.9%, respectively), although the fusion rate was excellent in any case [36]. A lower fusion rate was also observed in PEEK + ABG cages compared to PEEK + rhBMP2 cages (85.3% vs. 98.6%) [37]. PEEK + Ti alloy cages showed an improved fusion rate and MCS and lower VAS over time [38]. Non-Plasmapore-coated and Plasmapore-coated Ti cages experienced a similar increase in solid fusion rate over time, with no statistically significant differences (86% in the non-Plasmapore-coated Ti group vs. 100% with the Plasmapore-coated Ti group) [39].

### 3.7. Risk of Bias Assessment

Table 3 shows the Coleman scores for all the studies. The total score ranged from 49 [22,30] to 83 [32], and the studies were prospective cohort, controlled or uncontrolled or retrospective studies, open, single-center, prospective, single-arm phase I/II clinical or randomized clinical pilot trials. Most of the studies enrolled between 51 and 100 patients, followed by studies that enrolled more than 100 or 30–50 patients. Almost all of the studies had an f-up between 12 and 36 months, except for one that did not specify the f-up period [28] and one that had a f-up period greater than 61 months [33]. The single approach was the most used, and most of the studies described patient diagnosis with percentages, with adequate description of surgical technique but without a description of post-operative rehabilitation. In all but one study [26], the outcome measures, outcome timing, and reliable and general health measures were clearly defined. Regarding the procedure for assessing outcomes, in none of the studies were the outcomes assessed by patients themselves with minimal investigator assistance, and in some studies, the investigator was not independent of the surgeon. In one study, written consent was not specified [33], and in eight studies, only information on patient recruitment was included. Lastly, regarding the subject selection process, most of the studies reported patient selection criteria, but the recruitment rate was lower than 90%. The other studies described patient selection, and the recruitment rate was higher than 90%, and a few studies did not describe selection criteria, and the recruitment rate was lower than 90%.

### 3.8. Representative Cases of Complications from Our Institution

We report here two cases of complications concerning lumbar cages in transforaminal lumbar interbody fusion (TLIF) from our surgical experience. The first case is a 51-year-old female patient who underwent a revision surgery at our institution following herniectomy and the positioning of an interspinous device performed at another hospital. During the revision surgery, the interspinous device was removed, posterior arthrodesis L4-L5 was performed, and a carbon cage was inserted with TLIF in the L4-L5 intervertebral space after decompression, discectomy L4-L5 and preparation of vertebral plates. No complications were detected intraoperatively or post-operatively (Figure 3A). During the follow up, the patient complained of pain, and signs of pseudoarthrosis were observed on CT scan and MRI. In particular, 12 months after surgery, pseudoarthrosis with cage subsidence was evident on CT scan (Figure 3B).

The second case is a 69-year-old man affected by spondylolisthesis L4L5 grade I with severe stenosis and previous L3 fracture. A posterior arthrodesis L4-L5 was performed, and a titanium cage was inserted with TLIF after decompression, discectomy L4-L5 and preparation of vertebral plates. Then, vertebroplasty was performed in L3 with prophylactic aim. No complications were detected postoperatively. In a postoperative CT scan, the cage appeared slightly sunken on the inferior surface (Figure 4A), and after 3 months, the cage subsidence was evident in the superior surface (Figure 4B). We suggest that the initial slipping of the case was due to suboptimal preparation of the intervertebral space, while the significant cage subsidence observed at the 3-month follow up could be associated with poor bone quality due to previous treatments (chemotherapy and radiotherapy) for Hodgkin lymphoma.

## 4. Discussion

This systematic review collected clinical studies from the last 10 years that evaluated the use of cages in different SF surgical procedures performed for spine pathologies, and we focused on peri-operative and post-operative complications.

SF procedures have greatly increased in the last few years [40] after conservative treatment fails for several different spinal diseases at all spinal levels; however, few clinical studies report complications.

In this systematic review, we studied lumbar SF (9/21 studies) comprising PLF or PLIF, TLIF, ALIF and XLIF. Since the 1950s, posterolateral fusion and interbody fusion achieved significant clinical results in lower back pain treatment and were usually used to treat various spinal disorders, such as disc herniation, stenosis or spondylolisthesis, deformity, trauma and DDD, that are the most common causes of disability and chronic lower back pain, a relevant problem not only for patient health and quality of life, but also for healthcare costs [41,42].

The second largest category of spinal surgeries performed was ACDF (12/21 studies), which also started to be applied in surgery since the 1950s. It is usually employed for degeneration of the cervical spine and for spinal cord decompression to increase disc space, height and stiffness and to relieve neck pain, motor and sensory dysfunction, and radiculopathy or myelopathy that result in pain and weakness in the arms and poor quality of life for patients [43,44].

In SF, several different bone graft procedures amount to a total of nearly 2.2 million of these operations worldwide, with autograft as the gold standard [45]. Autograft from the iliac crest has a fusion rate of 90% and shows osteogenic, osteoinductive and osteoconductive abilities. However, due to some limitations in the use of autografts, such as donor site pain, wound infection, hematoma, pseudoarthrosis, bleeding and subsidence, other grafts (natural or synthetic) started to be used as alternatives [8]. Among autograft substitutes, allografts, DBM, ceramics, metal or plastic biomaterials gave support to SF procedures, reducing autograft complications and also increasing biomechanical performance. Recombinant proteins have also been combined with these biomaterials. In this review, autografts were obtained from local bone derived from laminectomy, osteophytes and the iliac crest.

In addition, PEEK was the most-employed biomaterial used to fabricate cages in lumbar SF and in ACDF (20/21 studies). It is a plastic cage, available in clinics starting in the 1990s, with biomechanical properties (elastic modulus and stiffness) similar to those of bone, and with good biocompatibility, even if a mild fibrous reaction has been reported [46]. It restores disc height alignment and reduces post-operative immobilization [45]. Most frequently, PEEK cages are also filled with autografts harvested from osteophytes, cancellous bone from the anterior iliac crest or morselized bone from the local decompression, with synthetic βTCP with or without HA, with rhBMP-2, with Ti coating or combined with Ti alloy, and/or with CS/DBM pellet or polybone.

Metals, in particular Ti alloys, were used first in cage fabrication, starting in the 1940s, due to their biocompatibility, robustness, corrosion resistance and low density [45]. In this review, Ti alloy was used in five studies. Ti alloy cages were enriched with local bone, standard Ti alloy was employed to make a combined instrumentation where the dynamic part was made of PEEK, or it was used to coat PEEK cages. Ti alloy endplate was combined with a PEEK body and allograft, and solid Ti alloy spacers were coated with a layer of fine Ti alloy powder. In clinics nowadays, it seems that PEEK has replaced Ti because PEEK is a material that does not induce allergies and possesses lower stiffness.

Most studies compared two types of cages, and all of them showed good clinical results in terms of pain, disability and symptom reduction (VAS, ODI, NDI and JOA scores), increased functional outcomes (SF-36 PCS, COMI score and PROLO score), disc height, fusion rate and quality of life (EQ-5D).

As summarized above, this systematic review includes clinical studies that used mixed cage treatments, unlike the approach followed in other systematic reviews that focused on a single type of cage [2,17,47]. With this ambitious review, we have tried to broaden the overview and the panorama, looking for a possible connection between cages and complications, even if with difficulty. A recent review showed that Ti and Ti-coated PEEK cages in PLIF procedures showed a similar rate of subsidence but a higher rate of fusion compared to PEEK interbody cages [48].

Many of the analyzed studies did not find complications related to the cages used; however, in the authors’ opinion, pseudoarthrosis, for example, has a pathogenesis that cannot be completely separated from the choice of cage, as also suggested by Iunes et al. [49]. As shown in Figure 5, for lumbar SF procedures, the highest percentages of subsidence (11%) and pseudoarthrosis (50%) were observed with PEEK + rhBMP2 and βTCP, respectively. For ACDF procedures, PEEK + ABG showed the highest percentages of subsidence (25%) and pseudoarthrosis (13%).

In studies where PEEK cages were used, there were very low rates of dural tears, superficial wound infection, non-fusion with pseudoarthrosis, cage subsidence, delayed incision healing, radiculopathy, small bowel ileus, sympathetic chain injury, atelectasis, hematomas, prolonged pseudo-obstruction of the colon, deep venous thrombosis (DVT), bilateral pleural effusions, aspiration pneumonia, urinary tract infection (UTI), transient dysphagia, revision surgery at the adjacent level, incidental durotomies, recurrent symptomatic nerve root compression, transient hoarseness and disk herniation at the lower level.

The addition of ABG taken from local bone to the PEEK cage seemed to be safer than ABG from the iliac crest because of fewer complications at the donor site. Similarly, the addition of CS/DBM instead of ABG showed less operative blood loss and fewer complications at the donor site, while the addition of rhBMP-2 induced higher 30-day readmission and use of oral steroids than βTCP. Two studies showed that the combination of βTCP and PEEK was not recommended for PLIF due to the high rate of pseudoarthrosis, while it was safe in the ALIF procedure. The use of an implant made of a dynamic part of PEEK and silicon and pedicle titanium screws was associated with a high rate of implant failure and adjacent segment disease.

Broad inclusion criteria were adopted initially to identify and include all relevant studies about this topic, as suggested by the choice of non-stringent keywords. All available clinical studies were systematically screened, including RCTs and prospective and retrospective studies to collect evidence about the selected topic. However, as per all reviews based on clinical studies of different types and design, speculations are partly limited by the quality of study design and by systemic biases (i.e., selection bias, information bias during data collection and especially reporting bias connected to the outcomes as complications). Indeed, surprisingly, we did not find any precise indications about 3D-printed devices. Recent literature data show that this technology has several advantages in terms of biological and biomechanical properties of the devices that are able to significantly improve the performance and the final outcome [50]. Again, we did not retrieve data on personalized approaches, meaning use of devices fabricated based on the patient’s anatomical needs, but this type of approach is certainly less frequent for making interbody fusion cages. Another limit found in the included studies is related to the ERAS protocol, which is a multidisciplinary protocol for patient management, especially in the pre-operative phase, that significantly reduces the onset of complications. [51]. We did not even find references or evidence about the adoption of this protocol. Nevertheless, in the review, a wide time range was covered, and it is likely that in many articles the ERAS protocols as actually codified were not present, but in some papers the information reported seems to suggest a more multidisciplinary approach to the patient. Despite the different biases and limits, we tried to pool all the available information to provide the best update about complications associated with the cages used for SF.

## 5. Conclusions

To conclude, considering the wide range of cages used, especially in combination, it is difficult to establish whether complications such as pseudarthrosis and subsidence can be directly linked to the cages. For example, bone substitutes are very different from each other in their properties and physicochemical characteristics. The same use of autologous material, bone tissue but also BMA, also depends on the patient’s state of health—the presence of comorbidities can affect the quality and quantity of biological material (smoking habits, diabetes, steroid use, osteoporosis, poor nutrition and age). Moreover, the use of bone substitutes and biological adjuvants is difficult to standardize because it often depends on the clinical needs that become evident during surgery. Therefore, the extreme variability in many aspects related to SF procedures makes it very difficult to establish a possible connection between complications and cages used. However, looking at the summarized data in this review, it can be speculated that the choice of graft could probably be related to the development of pseudarthrosis. Most of the studies included in the review highlighted the role of surgical techniques in patient complications. Several interacting events contextually affect clinical success and failure rates.

In general from our experience, the possible causes of complications related to cages, in particular pseudoarthrosis and subsidence, could be: suboptimal preparation of the vertebral plates before cage insertion, poor bone quality, previous surgery at the same site, composition of the cage or composition of the graft, and surgical technique (anterior or posterior approach).

We use interbody cages made of different materials (titanium, carbon fiber, PEEK and tantalium), but we have not observed any significant difference in the complication rates between different materials. Concerning the type of graft, we cannot make any comparison because we generally use only autologous local bone graft.

Concerning the surgical technique, we generally adopt the posterior approach (PLIF/TLIF). However, we can hypothesize that anterior techniques (ALIF/XLIF) have a minor rate of pseudoarthrosis due to the higher surface for cage implantation (ALIF/XLIF) and the possibility of better preparation of the intervertebral space (ALIF).

## Figures and Tables

**Figure 1 jcm-11-06279-f001:**
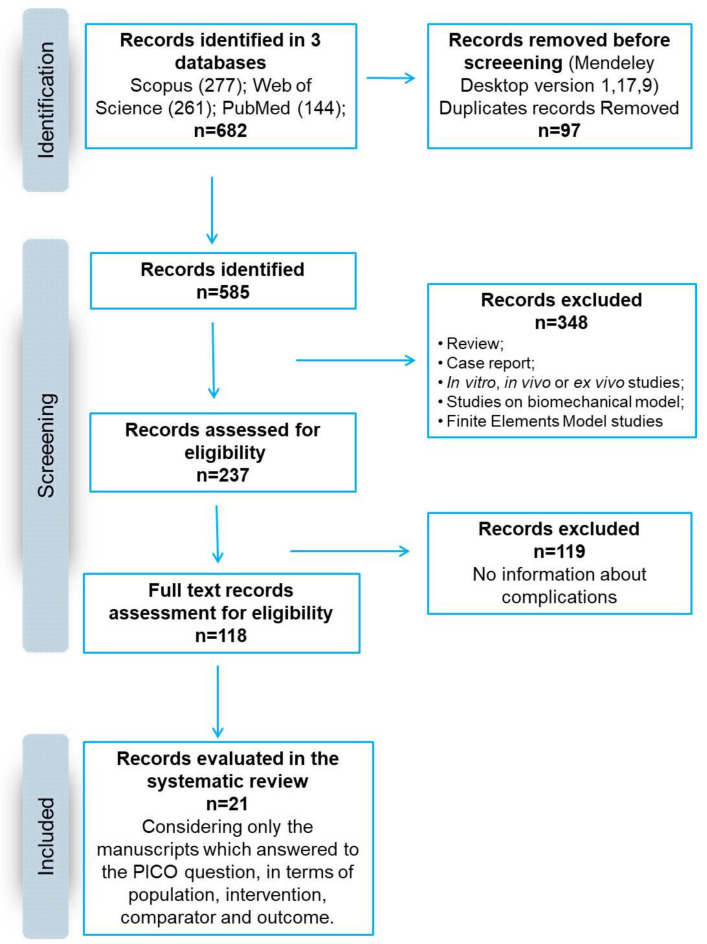
Schematic representation of the search strategy according to PRISMA principles.

**Figure 2 jcm-11-06279-f002:**
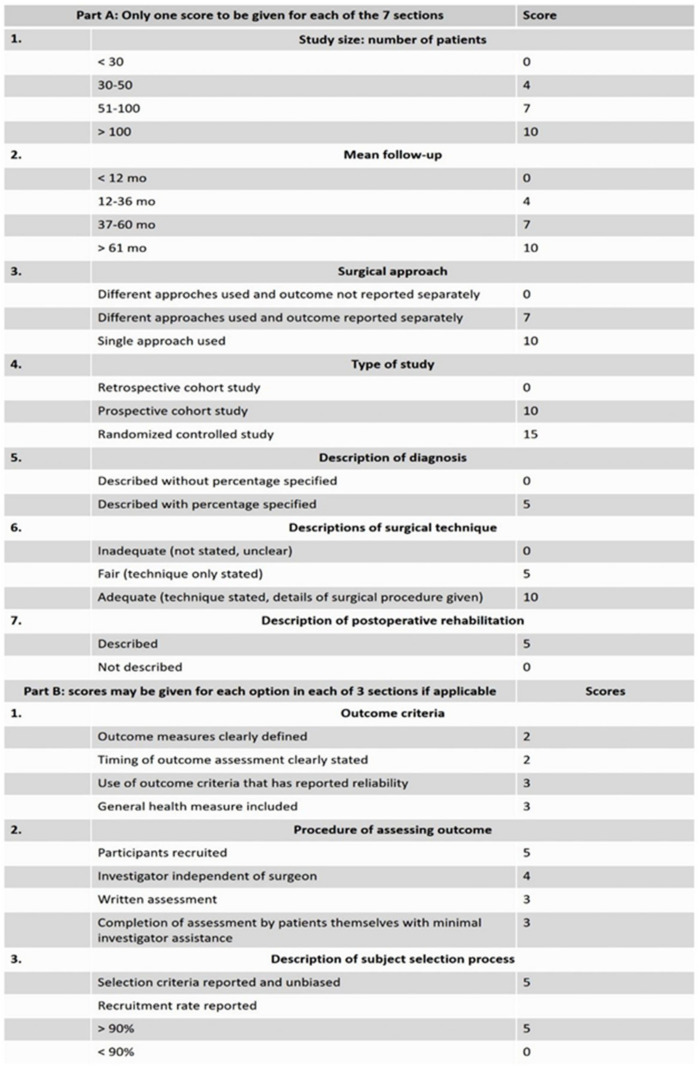
Coleman methodology score used for assessment of the risk of bias of the included articles.

**Figure 3 jcm-11-06279-f003:**
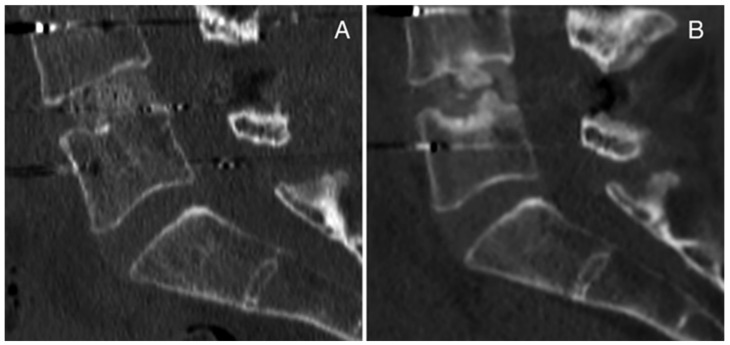
Case Report 1 from our institute: CT scan (**A**) post-operatively and (**B**) 12 months after surgery with pseudoarthrosis with cage subsidence.

**Figure 4 jcm-11-06279-f004:**
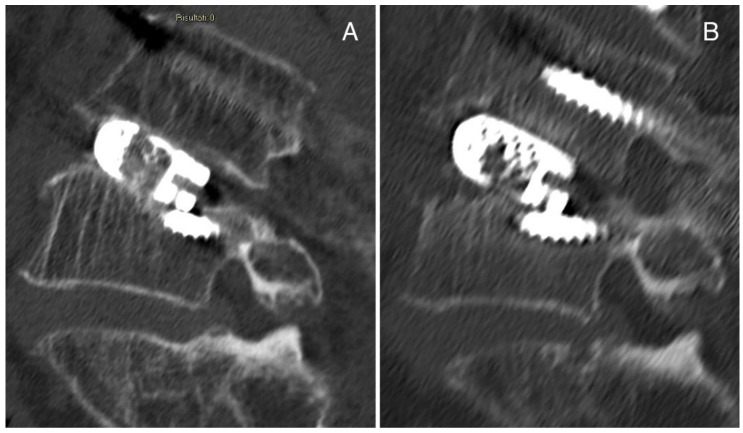
Case Report 2 from our institute: (**A**) postoperative CT scan in which the cage appears slightly sunken on the inferior surface, and (**B**) CT scan after 3 months in which the cage subsidence is evident in the superior surface.

**Figure 5 jcm-11-06279-f005:**
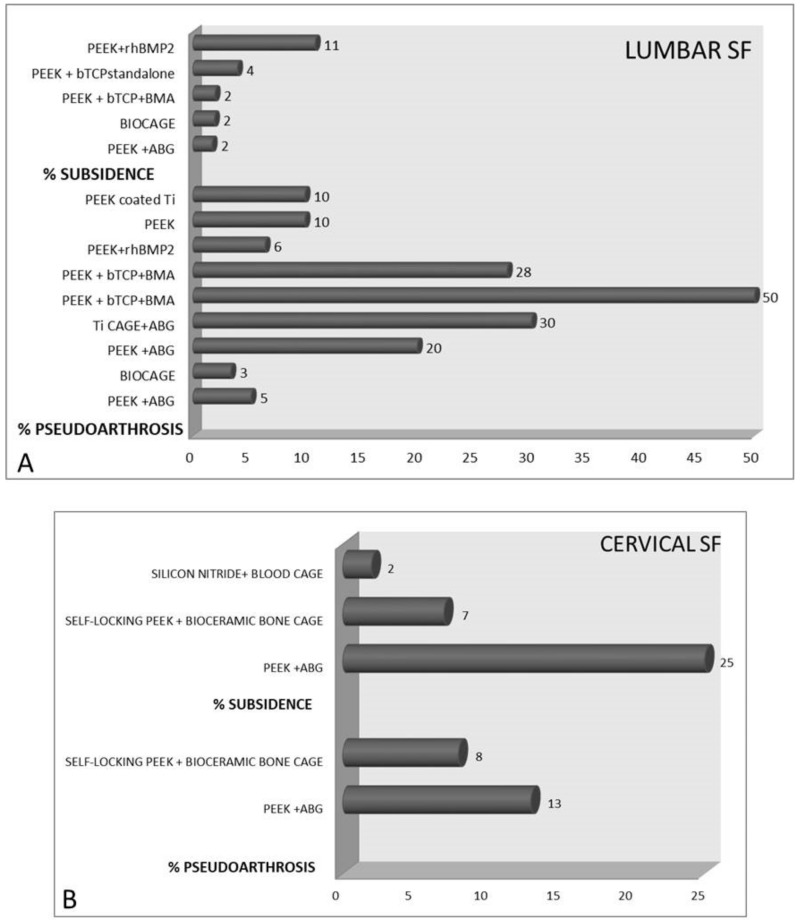
Bar charts that show the percentages of pseudoarthrosis and subsidence in patients treated with different cages for (**A**) lumbar spinal fusion and (**B**) anterior cervical discectomy and fusion. [PEEK = Polyetheretherketone; rhBMP-2 = recombinant human bone morphogenetic protein 2; βTCP = beta tricalcium phosphate; BMA = bone marrow aspirate; ABG = autologous bone graft; Ti = titanium].

**Table 1 jcm-11-06279-t001:** Inclusion and exclusion criteria of the search strategy.

Inclusion Criteria	- Types of papers: Clinical Study; Clinical Trial; Clinical Trial, Phase I, Phase II, Phase III, Phase IV; Comparative Study; Multicenter Study; Randomized; Prospective; Retrospective. - English language. - Patient who underwent spinal fusion procedure. - Description of the cages used. - Description of the complications observed in peri-operative and post-operative time.
Exclusion Criteria	- Types of Paper: Reviews; Case Reports; in vitro, in vivo or ex vivo Studies; Studies on Biomechanical Models; Finite Element Model Studies; Books; Chapters; Conference Proceedings; White Papers. - Non-English articles. - Procedures other than spinal fusion. - Without essential information about cages used. - Without essential information about complications.

**Table 2 jcm-11-06279-t002:** Clinical studies included in the systematic review. Summary of the most important aspects taken into consideration for the results description: type and follow-up of the study, cages used for spinal fusion surgery, complications encountered, main results, and references. The studies were also grouped according to the performed surgical procedure.

Study Type (f-up)	Cages (no. of pz)	Systemic and Local Complications	Fusion Results	Clinical Score Results	Ref.
** *Posterolateral lumbar interbody fusion (PLIF) 1-, 2- or 3-level PLIF* **					
Randomized study (24 mo)	PEEK cage (35 pz), ABG (34 pz)	PEEK cage and ABG: dural tears. PEEK cage: superficial wound infection, no cage loosening or breaking	PEEK cage and ABG: ↑ fusion rate, mean disc height	PEEK cage and ABG: ↓ pain, VAS score with good functional outcomes	[19]
Prospective, nonrandomized, controlled study (mean 32 mo)	PEEK cage + ABG (173 pz), Biocage (206 pz)	PEEK cage + ABG: similar operation time, blood loss, LOS, pseudoarthrosis, subsidence, delayed incision healing to Biocage	PEEK cage + ABG and Biocage: ↑ fusion rate. PEEK cage: ↓ mean height of intervertebral space recovery, height of intervertebral foramen recovery compared to Biocage	PEEK cage + ABG and Biocage: ↓ VAS, ODI	[20]
Prospective, uncontrolled study (12 mo)	PEEK + βTCP + BMA cage (34 pz)	Blood loss, transient paresis L5, dura leakage, migration of cage, seroma, inadequate fusion	↑ fusion rate	↓ ODI, VAS	[21]
** *Transforaminal lumbar interbody fusion (TLIF) 1-or multi-level TLIF* **					
Retrospective study (24 mo)	PEEK cage + ABG (40 pz), Ti cage + ABG (77 pz)	PEEK cage + ABG: pseudoarthrosis at 10 level. Ti cage + ABG: pseudoarthrosis at 16 level	PEEK cage + ABG: similar bone union rate to Ti cage + ABG	/	[22]
Observational, prospective, nonrandomized cohort study (24 mo)	PEEK + silicon cage added with Ti screw (22 pz)	LOS, pulmonary disease. Material failure in the dynamic portion, revision surgery, lumbar radiculopathy with no neurological deficit, misplaced pedicle screw and revision surgery, superficial wound infection, incidental durotomy	/	↓ COMI, VAS scores	[23]
Randomised controlled clinical pilot trial (12 mo)	PEEK cage (20 pz), TiPEEK cage (20 pz)	PEEK cage: similar revision for pseudoarthrosis, loose pedicle screws, i.o. hematoma to TiPEEK cage. TiPEEK cage: persistent leg pain, p.o. wound infection	PEEK and TiPEEK cages: ↑ fusion rate, preservation of disc height in the fused or adjacent segments	PEEK and TiPEEK cages: ↓ ODI score, ↑ EQ-5D. TiPEEK cage: ↑ VAS leg pain compared to PEEK cage	[24]
** *Anterior lumbar interbody fusion (ALIF) 1-, 2-, 3-level ALIF* **					
Prospective, randomized, controlled clinical trial (12 mo)	PEEK + βTCP + BMA cage (50 pz)	Blood loss, paresis L5, hematoma, vessel lesions, migration of cage, pseudoarthrosis, inadequate fusion anteriorly	↑ fusion	↓ ODI, VAS	[25]
Prospective, uncontrolled study (mean 12 mo)	PEEK + rhBMP-2 cage (131 pz)	Minor complications. Major complications, prolonged pseudo-obstruction of the colon, DVT, bilateral pleural effusions, aspiration pneumonia, UTI. Pseudoarthrosis	↑ interbody fusion	↓ ODI, VAS, ↑ SF-36 PCS and SF-36 MCS	[26]
** *Extreme lateral interbody fusion (XLIF)* **					
Retrospective study (24 mo)	PEEK + βTCP + HA cage (25 pz), PEEK + rhBMP-2 cage (110 pz)	PEEK + rhBMP-2 cage: hematoma. PEEK + βTCP + HA cage: similar radiculopathy, subsidence, superficial wound infection to PEEK + rhBMP-2 cage	PEEK + rhBMP-2 cage: ↑ fusion rate compared to PEEK + βTCP + HA cage	PEEK + βTCP + HA and PEEK + rhBMP-2 cages: ↓ ODI, VAS, ↑ SF-36 PCS, SF-36 MCS	[27]
** *Anterior cervical discectomy and fusion (ACDF) 1- and 2-level ACDF* **					
Retrospective analytical observational cohort study	PEEK + ABG cage (30 pz), ABG (30 pz)	PEEK + ABG cage: ↓ operation time compared to ABG. ABG: donor site chronic pain, surgical wound infection, reoperation due to broken fixation system screw	PEEK + ABG cage and ABG: ↑ fusion rates, recovery of disc space height. PEEK + ABG cage: similar fusion rate as ABG	PEEK + ABG cage and ABG: ↑ clinical results	[28]
/ (24 mo)	PEEK + ABG cage (29 pz), ABG (31 pz)	PEEK + ABG cage: ↓ operation time, blood loss, perioperative complications compared to ABG. PEEK + ABG cage: dysphagia. ABG: donor site pain, dysphagia, wound infections	PEEK + ABG cage and ABG: ↑ fusion rate. PEEK + ABG cage: similar fusion rate as ABG	PEEK + ABG cage and ABG: ↓ VAS, ↑ JOA score. PEEK + ABG cage: ↓ DSH than ABG	[29]
Prospective study (24 mo)	PEEK + ABG cage (28 pz)	4-level ACDF: ↑ operation time, bleeding compared to 2- and 3-level ACDF. 2-, 3- and 4-level ACDF: Transient dysphagia, subsidence. 3- and 4-level ACDF: significant dysphagia, pseudoarthrosis rate, transient donor site pain	2-, 3- and 4-level ACDF: ↑ solid fusion	2-, 3- and 4-level ACDF: ↓ VAS, excellent and good results	[30]
Retrospective study (mean 29 mo)	Self-locking PEEK + bioceramic artificial bone cage with or without plate fixation (54 pz)	Mild dysphagia. Mild pseudoarthrosis	With and without plate fixation: ↑ fusion rate	With and without plate fixation: ↑ NDI, JOA	[31]
Prospective, single-blind randomized controlled study (24 mo)	PEEK + ABG cage (48 pz), Silicon nitride + blood cage (52 pz)	PEEK + ABG cage: similar operation time, blood loss, LOS, transient dysphagia, subsidence, incidental durotomy, recurrent symptomatic nerve root compression as silicon nitride + blood cage. PEEK + ABG cage: ↓ revision surgery at the adjacent level than silicon nitride + blood cage	PEEK + ABG and silicon nitride + blood cages: ↑ fusion rate	PEEK + ABG and silicon nitride + blood cages: ↑ NDI, SF36, patient perceived recovery, ↓ VAS	[32]
Retrospective study (mean 96.4 mo)	PEEK + ABG cage (47 pz), nHA/PA66 + ABG cage (51 pz)	PEEK + ABG cage: similar wound infection, subsidence to nHA/PA66 + ABG cage	PEEK + ABG and nHA/PA66 + ABG cages: ↑ fusion rate, segmental lordosis. PEEK + ABG cage: similar fusion rate as nHA/PA66 + ABG cage	PEEK + ABG and nHA/PA66 + ABG cages: ↑ JOA score, ↓ VAS score, good clinical outcome.	[33]
Prospective, randomized, controlled clinical study (24 mo)	PEEK + ABG cage (33 pz), PEEK + CS/DBM cage (35 pz)	PEEK + ABG cage: ↑ operation time, blood loos, total complication rate compared to PEEK + CS/DBM cage. PEEK + ABG cage: similar LOS, minor complications, hoarseness, superficial wound infection as PEEK + CS/DBM cage	PEEK + ABG and PEEK + CS/DBM cages: ↑ fusion rate. PEEK + ABG cage: similar fusion rate as PEEK + CS/DBM cage	PEEK + ABG and PEEK + CS/DBM cages: ↓ VAS, ↑ JOA score	[34]
Retrospective study (mean 30 mo)	PEEK + ABG cage (23 pz), PEEK + PolyBone cage (24 pz)	PEEK + ABG cage: similar operation time as PEEK + PolyBone cage	PEEK + ABG and PEEK + PolyBone cages: ↑ fusion rate. PEEK + PolyBone cage: ↓ disc height, ↑ time taken for fusion compared to PEEK + ABG cage	PEEK + ABG and PEEK + PolyBone cages: ↓ NDI, NRS score	[35]
Prospective, single-blind, randomized, controlled clinical study (12 mo)	PEEK + βTCP cage (32 pz), Acrylic cage (32 pz)	PEEK + βTCP cage: similar transient hoarseness, new degenerative changes at each level of the cervical spine, disk herniation at lower level compared to acrylic cage	PEEK + βTCP cage: ↓ fusion rate, disc space height compared to acrylic cage. PEEK + βTCP cage: similar subsidence as acrylic cage	PEEK + βTCP cage: ↓ clinical outcomes compared to acrylic cage	[36]
Retrospective chart review (median 12 mo)	PEEK + βTCP cage (107 pz), PEEK + rhBMP2 cage (84 pz)	PEEK + βTCP cage: ↓ 30-day readmission, oral steroids compared to PEEK + rhBMP2 cage. PEEK + βTCP cage: similar LOS, postoperative neurologic deficit, any dysphagia, ICU asPEEK + rhBMP2 cage. PEEK + βTCP cage: hardware failures. PEEK + βTCP cage: ↑ subsequent cervical spine surgery compared to PEEK + rhBMP2 cage	PEEK + βTCP cage: ↓ fusion rate compared to PEEK + rhBMP2 cage	/	[37]
Prospective single senior surgeon cohort study (mean 14.6 mo)	PEEK + Ti alloy + allograft cage (24 pz)	Without anterior plate fixation: pseudoarthrosis	With and without anterior plate fixation: ↑ fusion rate	With and without anterior plate fixation: ↑ MCS, ↓ VAS, good and excellent clinical outcomes	[38]
Retrospective cohort study (mean 24 mo)	Non-Plasmapore-coated Ti cage (42 pz), Ti cage coated with Plasmapore (30 pz)	None-Plasmapore-coated Ti cage: similar blood loss, operation time as Ti cage coated with Plasmapore	Non-Plasmapore-coated Ti cage and Ti cage coated with Plasmapore: ↑ solid fusion rate	/	[39]

↑ = increased; ↓ = reduced; ABG = autologous bone graft; ACDF = anterior cervical discectomy and fusion; BMA = bone marrow aspirate; COMI = Core Outcome Measures Index; CS/DBM = calcium sulphate/demineralized bone matrix; DSH = disc space heights; DVT = deep venous thrombosis;; f-up = follow-up; HA = hydroxyapatite; ICU = intensive care unit; JOA = Japanese Orthopaedic Association; LOS = length of stay in hospital; MCS = Mental Health Component Summary; mo = month; NDI = Neck Disability Index; NRS = numeric rating scale; ODI = Oswestry dysfunction index; PA66 = polyamide 66; PEEK = Polyetheretherketone; pz = patients; Ref. = reference; rhBMP-2 = recombinant human bone morphogenetic protein 2; SF-36 MCS = Short Form 36 mental component summary; SF-36 PCS = Short Form 36 physical component summary; Ti = titanium; UTI = urinary tract infection; VAS = Visual Analog Scale; yrs = years; βTCP = beta tricalcium phosphate.

**Table 3 jcm-11-06279-t003:** Coleman score results. For each study, the Coleman score items were evaluated, and a final total score is provided.

	Part A							Part B			Total
	Study Size	Mean F-Up	Surgical Approach	Type of Study	Description of Diagnosis	Description of Surgical Technique	Description of p.o. Rehabilitation	Outcome Criteria	Procedure for Assessing Outcomes	Description of Subject Selection Process	
[19]	7	4	10	10	0	10	0	10	12	5	68
[20]	10	4	10	10	5	10	0	10	8	5	72
[21]	10	4	10	0	0	0	0	10	5	10	49
[22]	4	4	10	10	5	10	5	10	12	5	75
[23]	4	4	10	10	5	0	0	10	8	5	56
[24]	10	4	10	0	5	5	0	10	8	10	62
[25]	10	4	10	10	5	10	0	8	12	5	74
[26]	0	4	10	10	5	0	0	10	12	5	56
[27]	4	4	10	15	5	10	5	10	8	5	76
[28]	7	0	10	0	5	10	0	10	12	10	64
[29]	7	4	10	0	0	5	0	10	8	10	54
[30]	0	4	7	10	0	10	0	10	8	0	49
[31]	7	4	10	0	0	10	0	10	12	10	63
[32]	7	4	10	15	5	10	0	10	12	10	83
[33]	7	10	10	0	0	5	0	10	9	10	61
[34]	7	4	10	15	5	10	0	10	8	5	74
[35]	4	4	10	0	0	10	0	10	8	10	56
[36]	7	4	10	15	5	10	0	10	5	5	71
[37]	10	4	10	0	5	5	0	10	5	10	59
[38]	0	4	10	10	0	10	0	10	12	0	56
[39]	7	4	10	15	0	10	0	10	12	5	73

## Data Availability

Not applicable.

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
