# Peer review of "Complications in Spinal Fusion Surgery: A Systematic Review of Clinically Used Cages"

_jcm, 2022, doi:10.3390/jcm11216279_

Round 1
Reviewer 1 Report
Add a limitations section to the end of the discussion section.
Line 398: Please add a reference to the statement "SF procedures have greatly increased in the last few years..."
Line 426-428: Please add reference to support statement that cages began in the 1990s.
Edits:
Line 480: Change 'procedure' to 'procedures'
Line 157: 'there were 585 duplicate records...' No, this was the number of records left over after removing 97 duplicate records. Please correct.
Line 196: List three numbers with only two procedures? Please fix.
Line 200-201: List four numbers with only three procedures? Please fix.
Line 358: Change 'as regards the...' to 'as regards to the...'
Line 367: Change '51-years old...' to '51-year-old...'
Line 434: Change 'were the used first...' to 'were used first...'
Line 472: add 'and' prior to 'disk herniation...'
Author Response
Add a limitations section to the end of the discussion section.
We thank the Reviewer for gave us the opportunity to further implement our work. According to the Reviewer suggestion we modified the last paragraph of the Discussion section as follows, including the limits: “A broad inclusion criteria were adopted initially to identify and including all relevant studies about this topic as suggested by the choice of non-stringent keywords. All available clinical studies were systematically screened, including RCTs, prospective and retrospective studies to collect the evidence about the selected topic. However, as per all review, based on clinical studies of different types and design, speculations are partly limited by the quality of study design and by the systemic biases (i.e. selection bias, information bias during data collection and especially reporting bias connected to the outcomes as complications). Indeed, surprisingly, we did not find any precise indications about 3D-printed devices. Recent literature data show that this technology has several advantages in terms of biological and biomechanical properties of the devices, able to significantly improve the performance and the final outcome [51]. Again we didn’t retrieve data on a personalized approach, intended as use of devices made on the patient’s anatomical needs, but this type of approach is certainly less frequent for the making interbody fusion cages. Another limit found in the included studies, is related to the ERAS protocol, which is a multidisciplinary protocol for patient management especially in the pre-operative phase, that significantly reduce the onset of complications [52]. We did not even find references or evidence about the adoption of this protocol. Nevertheless, in the review a wide time range was covered, and probably in many articles the ERAS protocols as actually codified were not present, but in some papers the information reported seems to suggest a more multidisciplinary approach to the patient. Despite the different biases and limits, we tried to pool all the available information to provide the best up-dating about complications associated with the cages used for SF”.
Line 398: Please add a reference to the statement "SF procedures have greatly increased in the last few years..."
According to Reviewer suggestion, we added the following reference at number 40: “Reisener MJ, Pumberger M, Shue J, Girardi FP, Hughes AP. Trends in lumbar spinal fusion-a literature review. J Spine Surg. 2020 Dec;6(4):752-761. doi: 10.21037/jss-20-492”. The other refences were updated accordingly”.
Line 426-428: Please add reference to support statement that cages began in the 1990s.
According to Reviewer suggestion, we added the following reference at number 46: “Campbell, P. G., Cavanaugh, D. A., Nunley, P., Utter, P. A., Kerr, E., Wadhwa, R., & Stone, M. (2020). PEEK versus titanium cages in lateral lumbar interbody fusion: a comparative analysis of subsidence. Neurosurgical focus, 49(3), E10. https://doi.org/10.3171/2020.6.FOCUS20367”. The other refences were updated accordingly
Edits:
Line 480: Change 'procedure' to 'procedures'
Done
Line 157: 'there were 585 duplicate records...' No, this was the number of records left over after removing 97 duplicate records. Please correct.
We correct the sentence as follows: “There were 585 identified papers after duplicates removal (97 records) with Mendeley software”.
Line 196: List three numbers with only two procedures? Please fix.
We thanks the Reviewer for his/her suggestion and we corrected the text.
Line 200-201: List four numbers with only three procedures? Please fix.
We thanks the Reviewer for his/her suggestion and we corrected the text.
Line 358: Change 'as regards the...' to 'as regards to the...'
Done
Line 367: Change '51-years old...' to '51-year-old...'
Done
Line 434: Change 'were the used first...' to 'were used first...'
Done
Line 472: add 'and' prior to 'disk herniation...'
Done

Reviewer 2 Report
Since I am not an orthopaedic surgeon, but rather an academic with significant experience in reviewing systematic reviews I have restricted my comments to the formatting rather than the actual content. This is a well-researched and well-written paper which add greatly to the already published literature on this topic. However it is very long and not particularly reader friendly. My suggestion would be to move Tables 2 and 3 to a supplementary position and perhaps make a smaller summary table to be included in the body of the text.
Other formatting comments I have added to the paper - see attached.

Author Response
Since I am not an orthopaedic surgeon, but rather an academic with significant experience in reviewing systematic reviews I have restricted my comments to the formatting rather than the actual content. This is a well-researched and well-written paper which add greatly to the already published literature on this topic. However it is very long and not particularly reader friendly. My suggestion would be to move Tables 2 and 3 to a supplementary position and perhaps make a smaller summary table to be included in the body of the text.
We thank the Reviewer for gave us the opportunity to further implement our work and in agreement with his/her suggestion, a more concise and readable new Table 2 has been created, merging the information from previous Table 2 and 3, which have been removed by the paper.
Other formatting comments I have added to the paper - see attached.
We thank again the Reviewer for his/her work in improving our manuscript from the formatting point of view. We correct the manuscripts acoording to all Reviewer suggestions as you can see in the sumitted revised Papers
